# The Role of Antihyperglycemic Drugs and Diet on Erectile Function: Results from a Perspective Study on a Population with Prediabetes and Diabetes

**DOI:** 10.3390/jcm11123382

**Published:** 2022-06-13

**Authors:** Giuseppe Defeudis, Alfonso Maria Di Tommaso, Claudia Di Rosa, Danilo Cimadomo, Yeganeh Manon Khazrai, Antongiulio Faggiano, Raffaele Ivan Cincione, Nicola Napoli, Rossella Mazzilli

**Affiliations:** 1Unit of Endocrinology and Diabetes, Department of Medicine, University Campus Bio-Medico di Roma, 00128 Rome, Italy; a.ditommaso@unicampus.it (A.M.D.T.); n.napoli@policlinicocampus.it (N.N.); 2Unit of Food Science and Nutrition, Department of Science and Technology for Humans and the Environment, Campus Bio-Medico University of Rome, 00128 Rome, Italy; c.dirosa@unicampus.it (C.D.R.); m.khazrai@policlinicocampus.it (Y.M.K.); 3Clinica Valle Giulia, GeneraLife IVF Center, 00197 Rome, Italy; cimadomo@generalifeitalia.it; 4Unit of Endocrinology, Department of Clinical and Molecular Medicine, Sapienza University of Rome, 00185 Rome, Italy; antongiulio.faggiano@uniroma1.it (A.F.); rossella.mazzilli@uniroma1.it (R.M.); 5Department of Clinical and Experimental Medicine, University of Foggia, 71122 Foggia, Italy; ivan.cincione@unifg.it

**Keywords:** erectile dysfunction, sexual function, diabetes mellitus (DM), antihyperglycemic drugs, diet, glucagon-like peptide-1 receptor agonists (GLP-1a), metformin

## Abstract

Background. The purpose of this study was to evaluate the effect of diet and antihyperglycemic drugs on erectile dysfunction (ED) in a setting of subjects affected by diabetes mellitus (DM) or preDM. Methods. This is a prospective observational study on 163 consecutive subjects with preDM or DM. All patients have undergone a medical evaluation (age, Body Mass Index (BMI), family history of DM, duration of DM, smoking, physical activity, dyslipidemia, cardiovascular comorbidities, and testosterone and HbA1c levels) and the International Index of Erectile Function (IIEF)-5 questionnaire. Results. Overall, the mean age was 62.8 ± 9.3 years, and the mean BMI was 28.4 ± 4.6 kg/m^2^. The IIEF-5 score mean value was 14.4 ± 6.2 (range 4–25). Among all confounders investigated for their association with the IIEF-5 score, only age and the duration of DM among diabetic patients showed a significant trend. The IIEF-5 score was higher in patients using GLP-1a compared to insulin (16.7 ± 4.7 vs. 12.9 ± 6.2; *p* = 0.02). This association was confirmed after adjustment for age and duration of DM (*p* = 0.01). All other treatments were similar (14.9 ± 6.2, 14.8 ± 9.2, 15.3 ± 5.4, and 13.6 ± 6.8 for metformin, sulfonylureas (SU), dipeptidyl-peptidase-4 inhibitors (DPP-4i), and sodium-glucose cotransporter-2 inhibitors (SGLT2i) treatment, respectively). Conclusions. This prospective observational study increases attention and focus on the effect of antihyperglycemic drugs and diet on ED, above all about the role of new classes, showing a significant higher IIEF-5 mean value in patients using GLP-1a compared to patients on insulin treatment.

## 1. Introduction

Erectile dysfunction (ED), defined as the inability to achieve or maintain penile erection sufficient to obtain satisfactory sexual activity [1,2,3], can be due to many factors, among which diabetes mellitus (DM) play an important role [4,5,6,7]. DM is one of the most common chronic diseases due to impaired carbohydrate metabolism; alongside diet or nutritional supplements [8,9], educational therapy [10], metformin, sulfonylureas (SU), thiazolinediones (TZD), and insulin therapy, new antihyperglycemic drugs are now widely used for the management of DM, such as dipeptidyl-peptidase-4 inhibitors (DPP-4i), glucagon-like peptide-1receptor agonists (GLP-1a), and sodium-glucose cotransporter-2 inhibitors (SGLT2i) [11].

Considering the effect of antidiabetic treatment on erectile function (EF), a limited number of studies focused on humans are available [12,13,14], so the data are still limited on ED. A study conducted on Metformin use in diabetic men with ED showed an increased International Index of Erectile Function (IIEF) score after treatment [15]. Among the new antihyperglycemic treatments, GLP-1a showed promising effects on EF [16,17], by improving glycemic control and weight loss, while DPP4i and SGLT-2i needed more studies to draw conclusions [14,18,19]. About diet, the Mediterranean diet specifically resulted in effectively preserving EF, while Western diets could be a risk factor for ED [20,21]. The effects of diet of EF are related both to weight loss in men with obesity and T2DM [22,23] as well as to the type of food. Specifically, diets rich in plant foods, mainly the Mediterranean diet, are characterized by a high intake of polyphenols and antioxidants, which could increase nitric oxide (NO) availability [24].

The aim of this work was to evaluate the effect of diet and antihyperglycemic drugs at the time of recruitment on ED, considering both old and new therapeutic approaches, in a setting of patients affected by prediabetes or DM.

## 2. Materials and Methods

In this prospective observational study, 167 consecutive patients were enrolled from November 2021 to March 2022. All patients were admitted to the Department of Endocrinology and Diabetology of the University Campus Bio-Medico of Rome because they were affected by DM or prediabetes. Inclusion criteria were: (1) male sex; (2) age between 18 and 75 years; and (3) DM or prediabetes. At recruitment, patients underwent (a) a medical history evaluation; (b) anthropometric measurements such as Body Mass Index (BMI) calculated with formula: weight in kilograms/height in metres^2^ (BMI categories: normal weight < 25 kg/m^2^, overweight 25–29.9 kg/m^2^, obesity > 30 kg/m^2^); and (c) hormonal and biochemical assays. In addition, anamnestic forms were administrated by physicians and answered by all participants. The forms included: (a) personal information (date and place of birth), (b) personal and family history (parents or siblings) of DM; (c) lifestyle habits, including cigarette smoking (not/yes/in the past) and physical activity (no/yes, evaluated as at least 150 min/week of moderate-intensity activities, as reported by the Association of Diabetologists and Italian Association of Diabetology (AMD-SID) Guidelines [25]); (d) duration of DM; (e) cardiovascular (CV) comorbidities (CV events and hypertension); (f) dyslipidemia (not/yes); (g) diet (no/yes); and (h) antihyperglycemic drugs (metformin, DPP4i, SGLT-2, GLP-1a, insulin, or SU). The dietary scheme was indicated by nutritionists, in accordance with AMD-SID [25] and the American Diabetes Association (ADA) [26] guidelines, following the Mediterranean balanced diet. Specifically, the Mediterranean diet provides (a) carbohydrates < 130 g/day; (b) protein 10%–20% total calories intake; (c) total fat 20–35% of total calories intake, mainly polyunsaturated and monounsaturated fats; (d) sodium consumption of < 2.3 mg/day; and (e) alcohol not more than 20 g per day.

The choice of the DM or preDM pharmacological treatment was made according to the characteristics of the patient (age, glycemic compensation, and comorbidities), according to AMD-SID [25] and ADA guidelines [11].

Moreover, blood samples were obtained at 8:00 am and the plasma levels of total testosterone, glycosylated hemoglobin (HbA1c), triglycerides, high-density lipoprotein (HDL), and low-density lipoprotein (LDL) were measured. Chemiluminescence microparticle immunoassay (CMIA) and immunoassay (CLIA) were assessed. To evaluate EF, all patients completed the International Index of Erectile Function (IIEF)-5 questionnaire, a validated tool to outline ED (score: no ED > 21, mild ED 17–21, mild to moderate ED 12–16, moderate ED 8–11, severe ED 5–7) [27]. The IIEF-5 questionnaire was self-administered and fulfilled individually by pen and paper by each participant in a dedicated hospital room, without any possible influence and/or interference from physicians or other healthcare professionals. The primary outcome was to assess IIEF-5 according to the antidiabetic treatment administrated to each patient, namely diet, metformin, DPP4i, SGLT-2, GLP-1a, insulin, or SU. Patients in each specific group were on treatment for at least 6 months. Patients receiving metformin plus one of the other treatments were clustered according to the second one. Patients receiving insulin plus one or more than one of the other treatments were clustered in the insulin group. Patients referring to no regular sexual intercourse in the past six months, as required by the IIEF-5 questionnaire [27], were excluded.

The study was conducted in respect of the ethical standards of the Declaration of Helsinki (2000). The study was approved by the Institutional Ethics Committee of the University Campus Bio-Medico (protocol no. 0207655).

### Statistical Analysis

Continuous data were described as mean ± standard deviation (SD) with minimum and maximum values. Categorical data were described as absolute number and percentages. *T*-tests or ANOVA tests were adopted to assess statistically significant differences among continuous data. A Fisher exact test was performed to assess the differences among the categorical data. Moreover, linear regression analyses and generalized linear models were adopted to assess statistically significant associations between the antidiabetic treatment and IIEF-5 values. All data were adjusted for putative confounders investigated among all the features listed in the previous paragraph. *p*-values < 0.05 defined statistically significant differences. The software SPSS (IBM, Armonk, NY, USA) was used for the statistics. The investigator of the statistical analysis did not know the treatment regimen the patients took. We specified this data in the manuscript.

## 3. Results

A total of 167 patients DM or pre-DM were evaluated. Of them, four patients referred to no sexual intercourse in the past six months, thus, the final sample size was 163. No other patients refused to take part of the study. The main characteristics of these patients are summarized in Table 1. Overall, the mean age was 62.8 ± 9.3 years (range 20–75), and 90.8% of the patients were affected by DM (N = 148/163) and 9.2% by preDM (N = 15/163). The mean BMI was 28.4 ± 4.6 kg/m^2^ (range 18.4–45.9). A rate of 87.7% (143/163) of the patients referred to ED. The mean of the IIEF-5 score was 14.4 ± 6.2 (range 4–25). A total of 21 (12.9%) patients were solely on a diet, 51 (31.3%) were treated solely with metformin, and 4 (2.5%) were instead treated with SU. Considering new antihyperglycemic drugs, 19 (11.7%) patients were treated with DPP4i, 20 (12.3%) with GLP-1a, and 25 (15.3%) with SGLT-2. Lastly, 23 (14.1%) patients were treated with insulin; of them, 8 patients also received metformin plus SGLT-2 or DPP4i. No differences were observed in the prevalence of patients on a diet regimen among patients treated with antihyperglycemic drugs (40/51, 78.4%, 14/19, 73.7%, 16/25, 64.0%, 13/20, 65.0%, 16/23, 69.6%, and 4/4, 100%, for metformin, DPP4i, SGLT-2, GLP-1a, insulin, and SU, respectively (*p* = N.S.)).

When clustering the patients according to the antidiabetic treatment, all the parameters analyzed were compared through a *t*-test or Fisher exact test for continuous and categorical data, respectively (Table 1). The only significant differences reported were a higher prevalence of preDM in the diet group, compared to the total group, plus a shorter DM history together with lower HbA1c mean values in the diet group and a longer DM history together with higher HbA1c mean values in the insulin group, both compared to the total group (*p* < 0.05) (Table 1).

Among all patient features investigated (age, BMI, both considering BMI mean values and BMI categories, family history of DM, duration of DM, cigarette smoking, physical activity, cardiovascular comorbidities, dyslipidemia, testosterone levels, HbA1c, triglycerides, HDL, and LDL) for their association with the IIEF-5 score, only age (Linear regression analysis: constant = 21.1, 95%CI from 15.01 to 27.11, B = −0.1, 95%CI from −0.195 to −0.005, *p* = 0.04; generalized linear model: partial eta-squared = 0.026 [i.e., small association], *p* = 0.04) and the duration of DM among the diabetic patients (linear regression analysis: constant = 16.27, 95%CI from 14.73 to 17.80, B = −0.13, 95%CI from −0.242 to −0.018, *p* = 0.02; generalized linear model: partial eta-squared = 0.035 [i.e., small association], *p* = 0.02) showed a significant trend. Specifically, older patients and patients affected by diabetes for longer periods showed a progressively lower IIEF-5 score, as pictured in Figure 1A,B.

Across all treatment groups, the only significant difference in IIEF-5 score was a higher mean value in patients using GLP-1a compared to patients on insulin treatment (16.7 ± 4.7 vs. 12.9 ± 6.2; *p* = 0.02). This association was adjusted for age and duration of DM and confirmed a partial eta squared 0.16 (i.e., large association) with a *p* = 0.01 (Figure 2). Table 2 also reports the linear regression analysis. Furthermore, clustering the subgroup according to the IIEF-5 categories (no ED, mild ED, mild to moderate ED, moderate ED, and severe ED), a significant difference was observed in the prevalence of mild ED, which resulted as lower in the insulin group compared to GLP-1 (2/23, 8.7% vs. 8/20, 40.0% vs. *p* = 0.03) (Table 1). Interestingly, none of the patients treated with GLP-1 showed severe ED.

To rule out possible interferences related to the association of insulin with multiple drugs, a sub-analysis was performed: there were no significant differences between patients taking only insulin and patients treated with insulin plus metformin and DPP4i/SGLT2i (12.3 ± 6.0 vs. 14.1 ± 5.8; *p* = 0.5).

All other treatments resulted in comparable values (14.9 ± 6.2, 14.8 ± 9.2, 15.3 ± 5.4, and 13.6 ± 6.8 for metformin, SU, DPP4i, and SGLT2i treatment, respectively) (Figure 2).

Finally, the mean ± SD of the duration treatment for each group resulted comparable among all categories, except for metformin, which resulted in more time (4.5 ± 3.7 years) compared to GLP-1 (2.8 ± 2.5 years) and SGLT-2 (2.1 ± 1.4 years) (*p* = 0.001) (Table 1).

## 4. Discussion

Sexual dysfunctions are well recognized as complications of DM, more in males [1,2,4,5,6,7] than in females [28,29]. In this study, we evaluated a total of 163 patients with DM or preDM. Interestingly, the prevalence of ED in our population was 88.0%. This data was partially in agreement with previous data [2,30,31]. Specifically, Fedele et al. found a prevalence of ED in Type 1 DM (T1DM) of 26% and in Type 2 DM (T2DM) of 37% [30], while Kouidrat et al. described in a recent meta-analysis a prevalence of ED of 37.5% and 66.3% in subjects with T1DM and T2DM, respectively [31]. Anyways, our findings could be partially explained by the fact that in our population the average BMI is indicative of marked overweight, which is itself a risk factor for ED [32,33], and the prevalence of CV comorbidities was higher than 90%. In this regard, current evidence shows that central obesity is particularly associated with arteriogenic ED [33]. Furthermore, endothelial dysfunction is a common link between ED and hypertension (de Oliveira 2021), and the ED severity in DM patients seems to be related to age and hypertension [34].

Among all parameters analyzed for their association with the IIEF-5 score (age, BMI, family history of DM, duration of DM, cigarette smoking, physical activity, comorbidities, dyslipidemia, testosterone levels, HbA1c, triglycerides, HDL, and LDL), the results of the present study highlighted that older patients and patients affected by DM for longer periods showed a progressively lower IIEF-5 score. Specifically, for each increase of one year of age, there is a decrease of 0.1 in the IIEF-5 total score. It is well known that age is an important risk factor for sexual dysfunction, mainly ED, and the prevalence increases with age, ranging from 1%–10% in subjects with less than 40 years to 50%–100% in subjects older than 70 years [1,35].

Considering the antidiabetic treatment in patients using GLP-1a, the IIEF-5 score showed better results than all other treatments but achieved a significant difference only in comparison to insulin. This association was confirmed after adjusting for age and duration of DM. Interestingly, none of the patients treated with GLP-1 showed severe ED, and the prevalence of mild ED was significantly higher in the GLP-1 group compared to the insulin group. Animal studies highlighted that GLP-1Ra could improve EF in diabetic rats, by a protective role on corpus cavernosum endothelial cells [17,36]. With regard to studies on humans, Giagulli et al. performed a retrospective observational study on 43 patients with obesity, hypogonadism, and DM, who referred to a new onset of ED. All subjects received testosterone undecanoate i.m. 1000 mg every 12 weeks and metformin (2–3 g/day) for 12 months. In patients with HbA1c > 7.5%, Liraglutide 1.2 µg/day was added for an additional 12 months. All patients underwent an IIEF-15 questionnaire at baseline, after 12 months, and after 24 months. The authors found a significant improvement in the IIEF-5 score compared to the group receiving only testosterone and metformin [16]. Similarly, Bajaj et al. studied the EF in DM patients treated with Dulaglutide, assessed by the IIEF-15 questionnaire at baseline, after 2 years, after 5 years, and at the study end, by randomization. The authors found that Dulaglutide improved EF in a subgroup of patients with DM and cardiovascular disease [37]. The effect of insulin on sexual function that emerged from this study is difficult to compare to previous studies, which are mainly based on the comparison between types of administration (multi drug injection—MDI—vs. continuous subcutaneous insulin infusion—CSII) [38,39] and not between insulin and any other treatment.

The diet followed by the patients was almost on the basis of the Mediterranean diet, when followed. As known, the Mediterranean diet may be associated with an improvement of EF [14,21,40]. The role of diet, when associated with antidiabetic drugs, allows to explain the favorable trend, though not significant, on the IIEF-5 score, compared to other groups.

### Limits and Future Perspective

The main limitation of this study is the small number of patients treated with some class of antihyperglycemic drugs (11.7–15.3%), albeit far beyond the percentage in using the new categories reported by an Italian report on subjects with diabetes (2–6%) [41]. In particular, the new antihyperglycemic drugs (mainly SGLT-2 and GLP-1) were introduced more recently than the others, so this also explains the shorter duration of treatment that we found. This limitation led to speculation about the effects of these drugs on EF and the generalizability of our findings. Anyways, the strength of this study is represented by the fact that these data could highlight a gap of knowledge to be filled, to support a “tailor-made” therapy for subjects with DM and preDM. Of note, these data are not applicable to patients not affected by DM.

Further randomized, double-blinded, and longitudinal studies could better clarify all these aspects.

## 5. Conclusions

This prospective observational study increases the attention and focus on the effect of antihyperglycemic drugs and diet on EF, in subjects with diabetes or pre DM, above all regarding the role of new classes (DPP4i, GLP1a, SGLT2i), showing that there was a higher prevalence of preDM in the diet group compared to the total group, a shorter DM history together with lower HbA1c mean values in the diet group, and a longer DM history together with higher HbA1c mean values in the insulin group, both compared to the total group. Furthermore, about the IIEF-5 score, there was a significant higher mean value in patients using GLP-1a compared to patients on insulin treatment.

## Figures and Tables

**Figure 1 jcm-11-03382-f001:**
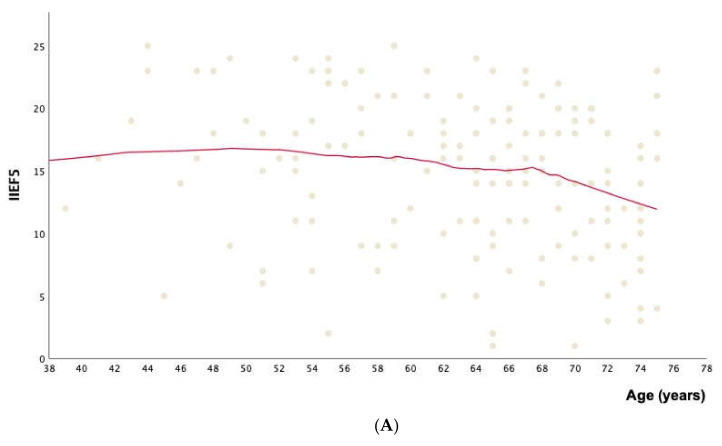
IIEF-5 score according to age (**A**) and duration of DM (**B**). Each dot in the dispersion plot in (**A**) identifies the International Index of Erectile Function (IIEF)-5 in each patient according to his age. The same outcome is reported according to the duration of diabetes in (**B**). The red lines represent the locally estimated scatterplot smoothing (LOESS) regression curve, with a fit to 50% of the points and an Epanechnikov weight function.

**Figure 2 jcm-11-03382-f002:**
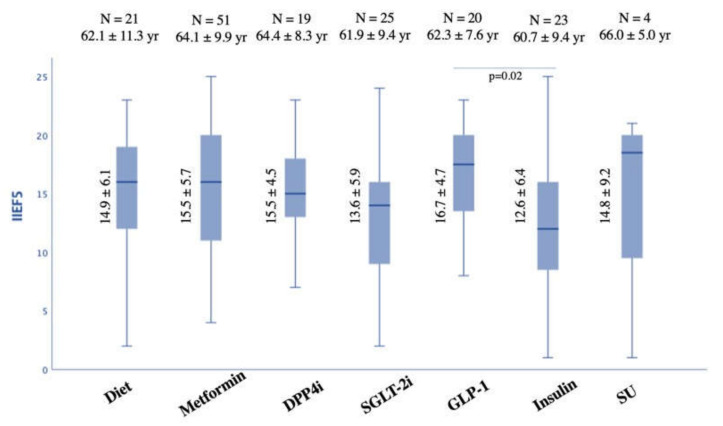
IIEF-5 score according to antidiabetic treatment group. *DPP4i = dipeptidyl-peptidase-4 inhibitors; GLP-1a = glucagon-like peptide-1receptor agonists; SGLT2i = sodium-glucose cotransporter-2; SU = sulfonylureas; IIEF = International Index of Erectile Function*.

**Table 1 jcm-11-03382-t001:** Basal characteristics of patients, in the total group and in subgroup according to antidiabetic treatment. *DPP4i = dipeptidyl-peptidase-4 inhibitors; GLP-1 = glucagon-like peptide-1receptor agonists; SGLT2i= sodium-glucose cotransporter-2 inhibitors; SU = sulfonylureas; IIEF = International Index of Erectile Function; DM = diabetes mellitus; BMI = body mass index; CV = cardiovascular. ^a^ p < 0.05 vs. total group. ^b^ p < 0.05 vs. insulin. ^c^ p < 0.05 vs. total group. ^d^ p < 0.05 vs. SGLT2i and GLP-1*.

	Total Group No. 163	Diet No. 21	Metformin No. 51	DPP4i No. 19	SGLT2i No. 25	GLP1a No. 20	Insulin No. 23	SU No. 4
**Age** (years; mean ± SD, range)	62.9 ± 9.3 20–75	62.1 ± 11.3 33–75	64.1 ± 9.9 20–75	64.4 ± 8.3 48–75	61.9 ± 9.4 41–74	62.3 ± 7.6 48–72	60.7 ± 9.4 39–75	66.0 ± 4.9 59–70
**preDM** (yes; no., %)	15/163 9.2%	15/21 71.4% ^a^	0/51 0%	0/19 0%	0/25 0%	0/20 0%	0/23 0%	0/4 0%
**IIEF-5** (mean ± SD)	14. ± 5.8	14.9 ± 6.1	15.5 ± 5.7	15.5 ± 4.5	13.2 ± 5.9	16.7 ± 4.7 ^b^	12.6 ± 6.4	14.8 ± 9.3
**IIEF-5 groups**								
No ED (> 21)	21 (12.8%)	2 (9.5%)	8 (15.7%)	2 (10.5%)	2 (8.0%)	4 (20.0%)	3 (13.0%)	0 (0%)
ED (5–21)	142(87.2%)	19 (90.5%)	43 (84.3%)	17 (89.5%)	23 (92.0%)	16 (80.0%)	20 (87.0%)	4 (100%)
*Mild ED (17–21)*	*48 (29.4%)*	*8 (38.1%)*	*16 (31.4%)*	*7 (36.8%)*	*4 (16.0%)*	*8 (40.0%) b*	2 (8.7%)	*3 (75.0%)*
*Mild/Moderate (12–16)*	*44 (27.0%)*	*6 (28.6%)*	*13 (25.5%)*	*6 (31.6%)*	*8 (32.0%)*	*4 (20.0%)*	7 (30.4%)	*0 (0%)*
*Moderate ED (8–11)*	*30 (18.4%)*	*2 (9.5%)*	*8 (15.7%)*	*3 (15.8%)*	*7 (28.0%)*	*4 (20.0%)*	6 (26.1%)	*0 (0%)*
*Severe ED (5–7)*	*20 (12.3%)*	*3 (14.3%*	*6 (11.8%)*	*1 (5.3%)*	*4 (16.0%)*	*0 (0%)*	5 (21.3%)	*1 (25.0%)*
**BMI**	28.4 ± 4.6	28.1 ± 3.5	28.1 ± 3.6	26.4 ± 3.1	29.8 ± 5.9	29.3 ± 5.6	28.1 ± 5.0	30.4 ± 2.1
(Kg/m2; mean ± SD, range)	18.4–45.9	19.6–36.7	19.6–36.7	18.4–32.7	21.1–44.3	22.5–45.9	19–38.6	28.1–33.0
**BMI groups** (n,%)								
Normal weight	36 (22.1%)	6 (28.6%)	8 (15.7%)	6 (31.6%)	5 (20.0%)	5 (25.0%)	6 (26.1%)	0 (0%)
Overweight	72 (44.2%)	9 (42.8%)	27 (52.9%)	10 (52.6)	10 (40.0%)	7 (35.0%)	7 (30.4%)	2 (50.0%)
Obese	55 (33.7%)	6 (28.6%)	16 (31.4%)	3 (15.8%)	10 (40.0%)	8 (40.0%)	10 (43.5%)	2 (50.0%)
**Physical activity**	64/163	8/21	22/51	7/19	8/25	2/20	14/23	3/4
(yes, no., %)	39.4%	38.1%	43.1%	36.8%	32.0%	10.0%	60.9% ^b^	60.9%
**Smoking habits**								
**No (no., %)**	53 (32.5%)	9 (42.9%)	23 (45.1%)	1 (5.3%)	6 (24.0%)	9 (45.0%)	3 (13.0%)	2 (50.0%)
**Yes (no., %)**	34 (20.9%)	4 (19.0%)	5 (9.8%)	7 (36.8%)	6 (24.0%)	2 (10.0%)	10 (43.5%) ^c^	0 (0%)
**In the past (no., %)**	76 (46.6%)	8 (38.1%)	23 (45.1%)	11 (57.9%)	13 (52.0%)	9 (45.0%)	10 (43.5%)	2 (50.0%)
**Family history of DM** (no., %)	54/163 74.2%	11/21 52.4%	38/51 74.5%	13/19 68.4%	19/25 76.0%	17/20 85.0%	20/23 87.0%	4/4 100%
**DM duration** (years; no., %)	10.9 ± 8.3 0–40	5.7 ± 5.1 0–15 ^c^	7.2 ± 6.6 0–30	13.2 ± 9.2 1–34	9.6 ± 6.6 0–25	12.9 ± 6.9 4–30	17.7 ± 8.9 2–40 ^c^	12.5 ± 11.9 4–30
**Treatment duration** (mean ± SD)	3.7 ± 3.1	2.0 ± 1.3	4.5 ± 3.7 ^d^	4.5 ± 2.9	2.1 ± 1.4	2.9 ± 2.4	5.1 ± 3.8	4.5 ± 2.9
**CV comorbidities** (no., %)	152 93.3%	19 90.5%	49 96.1%	17 89.5%	25 100%	19 95.0%	19 82.6%	4 100%
**Dyslipidemia** (no., %)	121 74.2%	14 66.7%	39 76.5%	16 84.2%	18 75.0%	14 70.0%	16 69.6%	4 100%
**HbA1c** (%; mean ± SD, range)	7.1 ± 1.3 5–16	6.0 ± 0.6 5.0–6.8 ^c^	6.5 ± 0.8 5.5–9.4	7.2 ± 0.8 6.0–9.0	7.3 ± 0.9 5.5–9.6	7.4 ± 0.8 5.8–8.3	8.2 ± 2.3 5.2–16.0 ^c^	6.4 ± 0.8 5.6–7.2
**Total Cholesterol** (mg/dL; mean ± SD, range)	160.2 ± 37.5 73–338	181.3 ± 47.0 140–338	157.2 ± 33.9 97–226	153.1 ± 42.4 73–242	165.4 ± 36.2 94–226	147.8 ± 31.1 89–214	166.3 ± 34.9 117–237	124.0 ± 36.8 98–150
**Triglycerides** (mg/dL; mean ± SD, range)	135.8 ± 69.1 23–388	124.3 ± 47.7 37–223	128.6 ± 61.3 50–353	136.2 ± 79.5 60–388	164.2 ± 86.9 52–379	139 ± 65.5 51–259	127.0 ± 74.2 23–283	121.0 ± 36.8 98–150
**HDL Cholesterol** (mg/dL; mean ± SD, range)	47.6 ± 12.4 23–105	51.5 ± 9.3 33–65	48.7 ± 12.0 26–80	47.8 ± 19.1 24–105	44.8 ± 8.4 31–65	45.3 ± 7.5 32–61	49.0 ± 14.8 27–81	38.5 ± 21.9 23–54
**LDL Cholesterol** (mg/dL; mean ± SD, range)	85.5 ± 36.1 14–238	114.9 ± 43.2 62.3–238.8	87.4 ± 32.9 14.8–157	81.3 ± 35.8 18.4–174	88.9 ± 39.6 37.8–145.6	77.1 ± 28.9 17.4–125	97.3 ± 38.3 38.0–178.2	61.3 ± 17.4 49–73.6
**Total Testosterone** (ng/mL; mean ± SD, range)	4.0 ± 1.8 0.3–11.5	4.0 ± 1.6 1.0–7.9	4.2 ± 1.7 1.7–8.8	4.6 ± 2.8 1.9–11.5	3.7 ± 1.7 1.5–7.1	3.3 ± 2.1 0.3–5.9	3.9 ± 1.5 0.9–5.6	2.6 ± 0.2 2.5–2.8 ^c^

**Table 2 jcm-11-03382-t002:** Linear regression analysis outlining the increase in IIEF-5 score among patients treated with GLP-1 vs. patients treated with insulin, adjusted for age and duration of diabetes mellitus (DM).

	B, 95%CI, *p*-Value
**Constant IIEF-5**	24.224, 95%CI from 11.665 to 36.783
**Age**	−0.218, 95%CI from −0.435 to −0.002, *p* = 0.048
**Duration of DM**	+0.091, 95%CI from -0.141 to +0.323, *p* = 0.413
**GLP-1 vs. Insulin**	+4.900, 95%CI from +1.256 to +8.544, *p* = 0.01

## Data Availability

Data are available after specific request to the authors.

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
