# Peer review of "The Role of Antihyperglycemic Drugs and Diet on Erectile Function: Results from a Perspective Study on a Population with Prediabetes and Diabetes"

_jcm, 2022, doi:10.3390/jcm11123382_

Round 1
Reviewer 1 Report
Dear Authors,
Thank you for the possibility to review this interesting manuscript, which is fluently written. The topic is of current interest and fits the scope as well as the section of the Journal. I would like to give some recommendations for revision before publication.
1. Title:
- Fits the scope, you might use erectile function instead of erectile dysfunction, to be more precise.
2. Abstract
-no comments
3. Introduction
-well elaborated, you might clarify the diet aspect, is the ED and diet more related with eating some type of food or losing weight
4. Methods
-please add some clarifications: is this study part of the larger study project; how many of patients refused to take part of the study; how many of the patients were not suitable; how the forms were filled in (pen and paper/ digital; individually/with help of someone)
-why you used IIEF-5 mean values instead of categories: no ED (22-25 points), mild (17-21) / mild to moderate (12-16) /moderate (8-11) /severe (5-7) ED recommended by Rosen and Rhoden?
-please add more information about the diet patients used; was the diet included also to other patient groups if yes how are you able to measure the differences
-how you measured physical activity and family history?
5. Results
-please add some information of the variety of involved participants (statistical test of the group differences), because the groups seems not to be similar (not only size of the group but probably also other aspect are different)
-please add information about ED prevalence (less than 22 points) to all medication groups
-please add information about BMI in categories (normal or overweight) prevalence to all medication groups
-please add information about how many years patients had used the medications (duration of treatment) and how did this influence the ED?
6. Discussion
- please cover the limits on the generalizability of your findings.
-what are the practical implications of the study?
7. References
-ref 35 is not available and needs to be more detail (is it report?)
8. Tables, figures
-fine
Author Response
Dear Authors,
Thank you for the possibility to review this interesting manuscript, which is fluently written. The topic is of current interest and fits the scope as well as the section of the Journal. I would like to give some recommendations for revision before publication.
- Title:
- Fits the scope, you might use erectile function instead of erectile dysfunction, to be more precise.
R: thank you, we modified the title as suggested.
- Abstract
-no comments
- Introduction
-well elaborated, you might clarify the diet aspect, is the ED and diet more related with eating some type of food or losing weight
R: Thank you. Following your suggestion, we added these sentences in the introduction: “The effects of diet of erectile function are related both to weight loss in men with obesity and T2DM (Esposito et al, Collins et al) and to the type of food. Specifically, diets rich in plant foods, mainly the Mediterranean diet, are characterized by high intake of polyphenols and antioxidants, which could increase nitric oxide (NO) availability (Adams et al).”
- Methods
-please add some clarifications: is this study part of the larger study project; how many of patients refused to take part of the study; how many of the patients were not suitable; how the forms were filled in (pen and paper/ digital; individually/with help of someone)
R: Only 4 patients were excluded, due to absence of regular sexual intercourse in the past six months. No other patients refused to take part of the study. We specified in the manuscript as follows: “A total of 167 patients with DM or pre-DM were evaluated. Of them, 4 patients referred no sexual intercourse in the past six months, thus the final sample size was 163. No other patients refused to take part of the study”.
We also specified in material and methods that: “The IIEF-5 questionnaire was self-administered and fulfilled individually by pen and paper by each participant in a dedicated hospital room without any possible influence and/or interference from physicians or other healthcare professionals”.
-why you used IIEF-5 mean values instead of categories: no ED (22-25 points), mild (17-21) / mild to moderate (12-16) /moderate (8-11) /severe (5-7) ED recommended by Rosen and Rhoden?
R: thank you very much for your suggestion. We have chosen the total score due to the small number of patients for each subgroup. Furthermore, these data allowed the comparison with other studies that used the score (see introduction/discussion, i.e. Giagulli et al, Rey-Valzacchi et al, ). However, we now added the results also according to IIEF5 categories. We specified in Materials and Methods as follows: “To evaluate EF, all patients completed the International Index of Erectile Function (IIEF)-5 questionnaire, a validated tool to outline erectile dysfunction (score: no ED >21, mild ED 17-21, mild to moderate ED 12-16, moderate ED 8-11, severe ED 5-7).”
We added all the values in the Table 1 and described the results as follows: “Furthermore, clustering the subgroup according to IIEF-5 categories (no ED, mild ED, mild to moderate ED, moderate ED and severe ED), a significant difference was observed in the prevalence of mild ED, which resulted lower in insulin group compared to GLP-1 (2/23, 8.7% vs 8/20, 40.0% vs p=0.03) (Table 1). Interestingly, none of the patients treated with GLP-1 showed severe ED.”
The power analysis is 60%, therefore we preferred avoiding a logistic regression analysis. We hope this is fine.
We also discussed these results in the Discussion section, as follows “Interestingly, none of the patients treated with GLP-1 showed severe ED, and the prevalence of mild ED was significantly higher in GLP-1 group compared to insulin.”
-please add more information about the diet patients used; was the diet included also to other patient groups if yes how are you able to measure the differences
R: Thank you for your suggestion. We provided to modify Materials and Methods section, improving the information about the diet patients used: “The dietary scheme was indicated by nutritionists, in accordance with AMD-SID and ADA Guidelines, following the Mediterranean balanced diet. Specifically, Mediterranean diet provides a) carbohydrates <130 g/day; b) protein 10–20% total calories intake; c) total fat 20–35% of total calories intake, mainly polyunsaturated and monounsaturated fats; d) sodium consumption to <2.3 mg/day, e) alcohol no more than 20 g per day.
We also specified in the results the prevalence of patients on diet regimen among patients treated with antihyperglycemic drugs: “No differences were observed in the prevalence of patients on diet regimen among patients treated with antihyperglycemic drugs (40/51, 78.4%, 14/19, 73.7%, 16/25, 64.0%, 13/20, 65.0%, 16/23, 69.6% and 4/4, 100%, in metformin, DPP4i, SGLT-2, GLP-1a, insulin, and SU, respectively (p=N.S.).”
- how you measured physical activity and family history?
R: thank you for this observation. We specified these aspects as follows: “personal and family history (parents or siblings) of DM”. “Physical activity (not/yes, evaluated as at least 150 minutes/week of moderate-intensity activities, as reported by ADA and SID-AMD guidelines)”.
- Results
-please add some information of the variety of involved participants (statistical test of the group differences), because the groups seems not to be similar (not only size of the group but probably also other aspect are different)
R: Thank you for your observation. We modified the following sentence in the results section: “When clustering the patients according to the antidiabetic treatment, all the parameters analyzed were compared through T-test or Fisher exact test for continuous and categorical data, respectively (Table 1). The only significant differences reported were a higher prevalence of preDM in Diet group compared to the total group, a shorter DM history together with lower HbA1c mean values in diet group and longer DM history together with higher HbA1c mean values in insulin group, both compared to the total group (p<0.05) (Table 1).”
-please add information about ED prevalence (less than 22 points) to all medication groups
R: we added the ED prevalence, as well as other IIEF-5 categories, for all medication group (See Table 1).
-please add information about BMI in categories (normal or overweight) prevalence to all medication groups
R: we added the BMI categories for all medication group (See Table 1).
-please add information about how many years patients had used the medications (duration of treatment) and how did this influence the ED?
R: All the patients were in treatment for at least six months. We added this information in Material and Methods section: “Patients in each specific group were on treatment for at least 6 months”. Furthermore, we evaluated the duration of treatment for each group, as follows: “Finally, the meanSD of the duration treatment for each group resulted comparable among all categories, except for metformin, which resulted longer (4.53.7 years) compared to GLP-1 (2.82.5 years) and SGLT-2 (2.11.4 years) (p=0.001) (Table 1).”
We also discussed this result, as follows: “In particular, the new antihyperglycemic drugs (mainly SGLT-2 and GLP-1), were introduced more recently than others, this also explains the shorter duration of treatment which we found.”
6. Discussion
- please cover the limits on the generalizability of your findings.
-what are the practical implications of the study?
R: Thank you very much for this suggestion. We added a section “Limits and future perspective” and we specified: “The main limitation of this study is the small number of patients treated with some class of antihyperglycemic drugs (11.7-15.3%), albeit far beyond the percentage in using the new categories reported by an Italian report on subjects with diabetes (2-6%) [35]. In particular, the new antihyperglycemic drugs (mainly SGLT-2 and GLP-1), were introduced more recently than others, this also explains the shorter duration of treatment which we found. This limitation led to speculate the effects of these drugs on EF and the generalizability of our findings. Anyway, the strength of this study is represented by the fact that these data could highlight a gap of knowledge to fill, to support a “tailor made” therapy for subjects with DM and preDM. Of note, these data are not applicable to patients not affected by diabetes. Further randomized, double blinded and longitudinal studies could better clarify all these aspects”
Reviewer 2 Report
The manuscript entitled “The role of antihyperglycemic drugs and diet on erectile dysfunction: results from a perspective study on a population with prediabetes and diabetes” investigated the effect of diet and antihyperglycemic drugs on DMED. The results are promising. However, some revisions are necessary.
1. Why didn’t the authors set a normal population of the same age as a control group?
2. How did the authors decide on the treatment options for their patients? How long was the treatment and how were the study endpoints determined?
3. Was the study double blinded? Did the investigator of the statistical analysis know the treatment regimen the patient took?
4. Were there differences in IIEF, blood glucose and other indicators of patients in each group before treatment? Whether the severity of erectile dysfunction before treatment in different groups is consistent?
5. Were there other disorders affecting erectile function in the included patients? Did the authors perform relevant analyses?
6. How is patient compliance? Was the frequency of sex included in the analysis?
Author Response
The manuscript entitled “The role of antihyperglycemic drugs and diet on erectile dysfunction: results from a perspective study on a population with prediabetes and diabetes” investigated the effect of diet and antihyperglycemic drugs on DMED. The results are promising. However, some revisions are necessary.
- Why didn’t the authors set a normal population of the same age as a control group?
R: thank you for your observation. We set up our study only on a population with diabetes or prediabetes, to evaluate the effects of individual therapies and avoid bias related to the presence/absence of disease, which is itself a risk factor of erectile dysfunction. Our results can be applied to diabetic patients, aiming at evaluating the association between ED and antidiabetic treatment. We could not generalize the results to non-diabetic patients.
For this reason, we added the following sentence in the discussion: “Of note, these data are not applicable to patients not affected by diabetes”
- How did the authors decide on the treatment options for their patients? How long was the treatment and how were the study endpoints determined?
R: Thank you for your kind suggestion. The choice of the DM or preDM treatment was made according to the characteristics of the patient (age, glycemic compensation, comorbidities), according to AMD-SID and ADA guidelines. We added this information in the manuscript, Materials and Methods section: “ The dietary scheme was indicated by nutritionists, in accordance with AMD-SID and ADA Guidelines, following the Mediterranean balanced diet. Specifically, Mediterranean diet provides a) carbohydrates <130 g/day; b) protein 10–20% total calories intake; c) total fat 20–35% of total calories intake, mainly polyunsaturated and monounsaturated fats; d) sodium consumption to <2.3 mg/day, e) alcohol no more than 20 g per day. The choice of the DM or preDM pharmacological treatment was made according to the characteristics of the patient (age, glycemic compensation, comorbidities), according to AMD-SID and ADA guidelines.”
All the patients were in treatment for at least six months. We added this information Material and Methods section: “Patients in each specific group had been on treatment for at least 6 months.
Furthermore, we added information regarding the duration of specific treatment, as follows: “Finally, the meanSD of the duration treatment for each group resulted comparable among all categories, except for metformin, which resulted longer (4.53.7 years) compared to GLP-1 (2.82.5 years) and SGLT-2 (2.11.4 years) (p=0.001) (Table 1).”
We also discussed this result, as follows: “In particular, the new antihyperglycemic drugs (mainly SGLT-2 and GLP-1), were introduced more recently than others, this also explains the shorter duration of treatment which we found.”
The endpoint of this study was the effect of antihyperglycemic drugs and diet on erectile function, assessed by IIEF-5 questionnaire, a validate tools for the diagnosis and the evaluation of the severity of erectile dysfunction.
- Was the study double blinded? Did the investigator of the statistical analysis know the treatment regimen the patient took?
R. Thank you for you for your observation. The investigator of the statistical analysis of this observational study did not know the treatment regimen the patient took. We specified this data in the manuscript, in the Statistical analysis section.
The study design is not “double blinded”, because the patients were all informed about the therapy they were taking, which was previously prescribed following the clinical practice and Guidelines (ADA and AMD-SID guidelines). We also added a section “Limits and future perspective”, and we specified that: “Further randomized, double blinded and longitudinal studies could better clarify all these aspects.”
- Were there differences in IIEF, blood glucose and other indicators of patients in each group before treatment? Whether the severity of erectile dysfunction before treatment in different groups is consistent?
R: Thank you. The purpose of our study was to assess the effect of diet and antihyperglycemic drugs on erectile function of patients with prediabetes o diabetes, taking a “picture” at the time of the visit. We did not include the "longitudinal study”, as many patients did not need to change their antidiabetic-treatment at the enrollment time. Furthermore, we have no data regarding patients' sexual function prior to the change of therapy, and the questionnaire is not validated for investigating sexual function prior to the past six months.Anyway, we are collecting data that will be the subject of future studies with this specific purpose.
We corrected the aim of the study as follows: “The aim of this work was to evaluate the effect of diet and antihyperglycemic drugs at the time of recruitment, both considering old and new therapeutic approaches, on ED in a setting of patients affected by prediabetes or DM.”
- Were there other disorders affecting erectile function in the included patients? Did the authors perform relevant analyses?
R: thank you for this suggestion. We analyzed cardiovascular comorbidities (including hypertension and previous cardiovascular events), testosterone level, HbA1c values and lipid profile. Now we added the presence/absence of dyslipidemia (see Table 1 and Material and Methods section). None of these parameters differs in the groups of patient’s treatments related.
- How is patient compliance? Was the frequency of sex included in the analysis?
R: Thanks for the observation. We included only patients who referred regular sexual intercourse in the past six months, as indicated in the IIEF-5 questionnaire.
We added the following information in the Material and Methods section: “Patients referring no regular sexual intercourse in the past six months, as required by the IIEF-5 questionnaire, were excluded”, as well as in the results: “A total of 167 patients DM or pre-DM were evaluated. Of them, 4 patients referred no sexual intercourse in the past six months, thus the final sample size was 163. No other patients refused to take part of the study.”
Round 2
Reviewer 1 Report
Thank you for the opportunity to review the revised manuscript. Thank you also for agreeing with the suggestions and all the revisions made.
Reviewer 2 Report
The authors have addressed and revised their manuscript according to the former comments. There are no more comments.